# Effect of conservation farming and biochar addition on soil organic carbon quality, nitrogen mineralization, and crop productivity in a light textured Acrisol in the sub-humid tropics

Jose Luis Munera-Echeverri[1]*, Vegard Martinsen[1]*, Line Tau Strand[1], Gerard Cornelissen[1,2], Jan Mulder[1]

**1** Faculty of Environmental Sciences and Natural Resource Management, Norwegian University of Life Sciences, Akershus, Ås, Norway, **2** Norwegian Geotechnical Institute (NGI), Ullevål Stadion, Oslo, Norway

\* jlmunera88@hotmail.com (JM); vegard.martinsen@nmbu.no (VM)

**Data Availability Statement:** All relevant data are within the paper and its Supporting Information files.

## Abstract

Conservation farming (CF), involving basin tillage, residue retention and crop rotation, combined with biochar may help to mitigate negative impacts of conventional agriculture. In this study, the effects of CF on the amount and quality of soil organic matter (SOM) and potential nitrogen (N) mineralization were investigated in a maize-soya-maize rotation in an Acrisol in Zambia. A large field was run under CF for 7 years and in the subsequent three growing seasons (2015–2018), four management practices were introduced to study effects on soil characteristics and crop yield. We tested i) a continuation of regular CF (CF-NORM) ii) CF without residue retention (CF-NO-RES); iii) Conventional (CONV), with full tillage and removal of residues; and iv) CF with 4 ton ha$^{-1}$ pigeon pea biochar inside basins and residue retention (CF-BC). The experiment involved the addition of fertilizer only to maize, while soya received none. Soya yield was significantly higher in CF systems than in CONV. Maize yields were not affected by the different management practices probably due to the ample fertilizer addition. CF-NORM had a higher stock of soil organic carbon (SOC), higher N mineralization rates, more hot-water extractable carbon (HWEC; labile SOC) and particulate organic matter (POM) inside basins compared to the surrounding soil (outside basins). Our results suggest that the input of roots inside basins are more effective increasing SOM and N mineralization, than the crop residues that are placed outside basins. CONV reduced both quality and quantity of SOM and N mineralization as compared to CF inside basins. CF-BC increased the amount of SOC as compared with CF-NORM, whereas N mineralization rate and HWEC remained unaffected. The results suggest benefits on yield of CF and none of biochar; larger impact of root biomass on the build-up of SOM than crop residues; and high stability of biochar in soil.

**Funding:** The study was funded by the Norwegian University of Life Sciences (NMBU) (https://www.nmbu.no/en) through PhD internal financing to Munera-Echeverri J.L and by the Faculty of Environmental Sciences and Nature Resource Management at NMBU (https://www.nmbu.no/en/faculty/mina) as part of the stipend to Vegard Martinsen and by the R&D project "Climate Smart Agriculture in Zambia (CSAZ): Soil benefits parameters research, conventional and conservation agriculture" funded by the Conservation Farming Unit (CFU) of Zambia (https://conservationagriculture.org/). The sponsors did not play any role in the study design, data collection and analysis, decision to publish, or preparation of the manuscript.

**Competing interests:** The authors have declared that no competing interests exist.

## Introduction

Soil organic matter (SOM) is important for agricultural and ecosystem services [1, 2]. SOM contributes to the mitigation and adaptation to climate change since it acts as a sink for $CO_2$, a major greenhouse gas, it helps storing plant nutrients [3] and it makes crop production more resilient to drought conditions by promoting soil aggregation [4] and water infiltration [5, 6]. In Sub-Saharan Africa (SSA) SOM depletion is one of the major causes of soil degradation [7]. Thus, there is a need for improved soil management alternatives that may contribute to increases in SOM.

Conservation farming (CF), a set of practices including i) minimum or no tillage ii) retention of crop residues and iii) crop rotation, has been suggested as a way to increase soil organic C (SOC) if its three principles are strictly applied [8–11]. CF in the form of planting basins has been promoted among small-scale farmers in countries like Zambia for more than two decades [12] with yield benefits for farmers if combined with complementary practices such as the use of improved crop varieties, adequate weed and pest control, among others [13, 14]. Besides, biochar, the C-rich pyrolysis product of agricultural waste, has been suggested as a way of sequestering C while improving soil fertility [15–18]. Soil amendment with biochar may alleviate soil acidity, improve soil water-holding capacity and prevent leaching of plant nutrients [19, 20], which all contribute to increased crop yields. Previous research in SSA showed that biochar addition inside planting basins in CF husbandry increased maize yield after one growing season [21, 22]. Nevertheless, there are no studies conducted over a larger number of seasons.

Most of the studies assessing the effect of CF on SOC have focused on the total amount of organic C, whereas fractions of SOC have been overlooked. Fractionation of SOC can be used to assess the quality of SOC, thus understanding the effect of soil management practices on the processes of decomposition and stabilization of SOM [23]. Among the multiple fractionation methods, particulate organic matter (POM; density lower than 1.6–2.0 g cm$^{-3}$; not strongly bound to minerals and composed mainly of partially decomposed fragments of roots and aboveground biomass) and hot-water extractable C (HWEC at 80 ˚C; dissolved organic C with a diameter $< 0.45\mu m$) have been shown to be sensitive to changes in soil management and can be used to isolate labile to intermediate fractions, which are expected to be affected most by CF practices [9, 24–26].

It has been shown that cultivation reduces the amount of POM in tropical and subtropical regions [24] due to increased decomposition and/or reduced input of biomass. Fortunately, biomass inputs may help restoring the initial amount of POM [27], being root biomass more effective than aboveground biomass [28]. In addition, HWEC has been found to be a good predictor of potential mineralizable N, which can be used to assess the capacity of soil to supply N [29] under different soil management practices [25]. To our knowledge, little research has been conducted on the effects of the principles of CF combined with biochar addition on N mineralization in SSA. Besides, biochar has been shown to impact SOC decomposition rates both positively and negatively, a process known as priming effect that may last from days to years [30, 31]. Priming effect in biochar research has been measured mainly as the changes of $CO_2$ efflux in incubation experiments, while measurements of N mineralization rate are scant in spite of being another known way of determining this phenomenon [32].

The present research builds on a previous assessment of the effects of seven years of CF using permanent planting basins (a hand-hoe based form of minimum tillage that is practiced by smallholder farmers in arid environments) in Zambian Acrisols on SOM. The results suggested an increase of SOC of about 2.9‰ yr$^{-1}$ and an increase of N mineralization rates inside basins as compared to the adjacent soil [33]. Root biomass and not residue retention was

suggested to be the reason for the changes in the amount and quality of SOC. Further assessment of the effect of alternative practices and residue retention is needed though. The present study involved the practices: i) normal CF, ii) CF without residue retention, iii) conventional tillage without retention of crop residue and iv) CF with biochar addition inside basins only.

The objectives were to a) measure crop yield and stover production under each treatment in a rain-fed maize-soya-maize rotation (2.5 years), b) to investigate the amount and quality of SOM and N mineralization rates in response to residue removal inside and outside basins under CF, while comparing the results with the conventional management, and c) to study SOM quality and N mineralization upon biochar addition inside planting basins.

## Materials and methods

### 2.1 Land use history and experimental setup

The experiment was done on a private farm (CENA farms; Mount Isabel), Mkushi, Zambia (S13˚45′25.7″ E29˚03′55.5″), with the permission of the owner of the land. No additional permissions were needed to conduct the field study. The permission for the transport of soil samples from Zambia to Norway was obtained via the University of Zambia (UNZA) and the Norwegian University of Life Sciences. The field studies did not involve endangered or protected species.

The study was conducted on a sandy loam (Acrisol) with an average annual precipitation of 1220 mm and a mean annual temperature of 20.4 ˚C [33, 34]. Prior to the experiment, the site had been under CF (planting basins and maize–ground nut rotation) for 7 years. For details about soil characteristics and soil management reference is made to Martinsen et al. [33]. In October 2015 four soil management practices were established in 24 m$^2$ plots, randomly distributed in 4 blocks (S1 and S2 Figs) Treatments included i) Conventional (CONV): full tillage to a depth of 10 cm and no residue retention. ii) CF Normal (CF-NORM): continuation of the practice implemented during 7 years prior to 2015, with basins fully opened before planting and crop residue added after harvest (viz. 4.4 ton ha$^{-1}$ maize stover and 2.9 ton ha$^{-1}$ soya stover in 2016–17) iii) CF no residue (CF-NO-RES): as CF-NORM but with residue removal after harvest iv) CF + Biochar (CF-BC): addition of 4 ton ha$^{-1}$ of pigeon pea biochar inside basins before planting in October 2015 only, while crop residues were applied as in CF-NORM. The planting basins were dug, using a hoe, to a depth of 20 cm, in agreement with the local practice of conservation farming; and biochar was added at a depth of 20 cm, mixed with the soil and subsequently, covered with more soil. Most of the biochar was placed at a depth of 8 to 20 cm. Seeds and fertilizer were mixed into the upper 8 cm of the basins. All CF plots had four rows of six planting basins (distance 80 cm, 90 cm between rows, basins were about 40 cm x 20 cm and 20 cm deep), that were planted with either three maize plants or 8 seeds of soya beans. The variety of maize was MRI 634. The conventionally tilled plots (CONV) had four rows with 18 (equally spaced) maize plants each, whereas for soya there were six planting stations per row at a distance of 80 cm in which 8 seeds per station were sown, following local practice.

The experiment was done for 3 growing seasons (2015–2018). Maize was planted in 2015/ 2016, soya in 2016/2017 and again maize in 2017/2018. Fertilizer was applied only to maize with no fertilizer addition to soya. All the soil management practices received the same amount of NPK fertilizer and urea per hectare. The fertilizer was applied differently in CF treatments and CONV. In all CF treatments, 17.1±0.8 g of NPK (N, P$_2$O$_5$, K$_2$O; 10-20-10) was applied per basin at pre-planting, corresponding to 237 kg ha$^{-1}$. Five and eight weeks after planting, top-dressing with urea (46:0:0) was applied, corresponding to 100 kg ha$^{-1}$ each time (200 kg ha$^{-1}$ Urea). In the CONV treatment, the NPK fertilizer was applied at emergence along the rows of

maize plants. Five and eight weeks after planting a topdressing with urea was applied (200 kg ha$^{-1}$). The application of N to maize crop corresponded to 116 kg N ha$^{-1}$ yr$^{-1}$.

## 2.2 Biochar

Biochar feedstock was pigeon pea (*Cajanus cajan*) stems, produced in a flame curtain kiln [35] at 600 ˚C. The chemical characteristics of the biochar were as described by Munera-Echeverri et al. [36] and include pH (10.4), Electrical Conductivity (1.4 mS cm$^{-1}$), acid neutralizing capacity (ANC$_{pH7}$; 49 cmol$_{(+)}$kg$^{-1}$), total organic carbon (56.1%), Total N (0.69%), total H (1.1%) and cation exchange capacity (6.6 cmol$_{(+)}$kg$^{-1}$).

## 2.3 Biomass production

Stover biomass and grain yield were measured immediately after harvest in all plots in each of the four blocks. The number of replicates was four. The values measured in the field were corrected for dry matter content. Root to shoot ratios of maize plants were measured in CF-BC, CF-NORM and CONV before harvest (Feb.2018; 12 weeks after planting) by digging the whole root system of 1 basin (3 plants) per plot in CF treatments and 3 neighboring plants in CONV. Root biomass was calculated by multiplying total aboveground biomass (grain plus stover) measured at the end of the growing season and the measured root to shoot ratio. Root depth and width was measured in CF-NORM and CF-BC.

## 2.4 Soil sampling

Soil samples were taken from all plots in each of the four blocks, at two depths. Soil was sampled in CF-NORM and CF-NO-RES outside and inside basins. In CONV, we sampled inside rows of maize plants and in between rows. CF-BC was sampled inside basins only. The first sampling campaign was carried out in May 2016 (end the growing season 2015–16, after harvest; CF-NORM and CF-BC) and the second in February 2018 (in the middle of the season 2017–18; all the treatments). Samples inside basins were taken by mixing one soil core of 2 cm diameter from five different basins (i.e. one bulked sample per treatment plot). Samples outside basins were taken from the area in between rows of basins in the same way. Samples were collected from 0 to 8 cm and from 8 to 20 cm depth. The samples were dried at 40 ˚C over a 7 days period before being sieved (2mm) at the Norwegian University of life Sciences (NMBU). The sample of one of the plots in CF-BC in 2018 was lost during transport from Zambia to Norway. In summary, the dependent soil variables in each treatment, either inside or outside basins, had four replicates per depth each year (2016 and 2018), except for CF-BC inside basins sampled in 2018, which had three replicates.

## 2.5 C and N analyses and stocks

Soil organic carbon (SOC) was analyzed using a TruSpec CHN analyzer (Leco Corporation). Since soil pH was below 6.5 (S1 Table) total C was used as a measure of SOC. Total N was analyzed by the Dumas method [37] using a TruSpec CHN analyzer (Leco Corporation). Stocks of organic C and N (ton ha$^{-1}$) were calculated by multiplying elemental concentration (inside and outside basins), bulk density (BD) and depth of sampling. C and N stocks in CF-BC outside basins was assumed to be the same as in CF-NORM. Bulk density was determined using 100 cm$^3$-steel rings at 5 to 10 cm in the season 2016–17 inside and outside basins in each treatment. The samples were transferred to plastic bags and dried at 105 ˚C at NMBU to determine dry matter content.

## 2.6 N mineralization

Nitrogen mineralization rates were determined in a 60-day incubation experiment at 20 ˚C. A total of 8 g of dried soil was added to 50 ml polypropylene tubes and the moisture content adjusted to 30% (v/v) by adding 1.8 ml distilled water. The lids were placed loosely on the tubes, allowing gas exchange. Water was replenished every 12–14 days after weighing. The tubes containing the samples at time 0 were frozen at -18 ˚C and stored until KCl extraction. Mineral N ($NH_4^+$ and $NO_3^-$) was determined before and after incubation, using 20 ml 2M KCl. Tubes were shaken horizontally during 1 hour at 200 strokes per minute and filtered using Whatman filter (589/3). Extracts were analyzed for $NH_4^+$ and $NO_3^-$ by Flow Injection Analysis (FIA tar 5010). Potential N mineralization rates were calculated by subtracting the initial amount of $NH_4^+$ and $NO_3^-$ at time 0, from the amount determined after 60 days. Net production of $NH_4^+$ and $NO_3^-$ (both in g $kg^{-1}$ $day^{-1}$) after 60 days were summed prior to calculation of potential N mineralization rates.

## 2.7 Hot-water extractable C (HWEC)

The HWEC was determined as described in [25]. In brief, 5 grams of dried soil and 30ml of deionized water were added to 50ml polypropylene tubes. The tubes were closed, shaken in vortex-shaker for one minute and placed in a laboratory water bath during 16 hours at 80 ˚C. Subsequently, the tubes were centrifuged at 1700 g and the supernatant was filtered using a 0.45 μm polyethersulfone filter. The samples were analyzed for dissolved organic carbon (DOC) using a total organic analyzer (TOC-V CPN, Shimadzu).

## 2.8 Density fractionation

Particulate organic matter can be either free (fPOM) or occluded (oPOM) in soil aggregates. Density fractionation was carried out following a modified method based on Leifeld & Kögel-Knabner (2005) [27] to determine POM, which was a combination of fPOM (wet sieved and density < 1.8 g $cm^{-3}$) and oPOM (ultrasonic dispersion with 22 J $ml^{-1}$, < 1.8 g $cm^{-3}$) fractions. In a pre-experiment, we found it difficult to differentiate between fPOM and oPOM as the amount of fPOM was very small. A sodium polytungstate ($Na_6H_2W_{12}O_{40}$) × $H_2O$) solution was prepared and its density adjusted to 1.8 g $cm^{-3}$. A total of 20 g dried and sieved soil (< 2mm) was weighted into 50ml polypropylene tubes and 30 ml of sodium polytungstate was added. The tubes were shaken gently 10 times end-over-end. Subsequently, the samples were dispersed ultrasonically with 22 J $ml^{-1}$ and centrifuged at 1700 g for 12 minutes. The supernatant with floating particles (fPOM and oPOM) was transferred onto a 20 μm sieve and rinsed with distilled water until the electrical conductivity dropped to < 100 μS $cm^{-1}$. The POM was dried at 105 ˚C for 24 hours and weighed. Next, the samples were milled and analyzed for total C and total N. The amount of POM relative to the bulk soil, the contribution of POC to SOC, as well as contribution of POM-N (PON) to total N was calculated.

## 2.9 Statistics

Statistical analyzes were conducted using R software [38]. We used one-way ANOVA to determine differences in crop biomass (Table 1) and C and N stocks between treatments. Two way ANOVA (treatment 3 levels and in vs. out 2 levels) was used to assess differences in SOC, total N, N mineralization rate and HWEC between treatments and between inside and outside basins (CF) or rows (CONV; Fig 1). Linear mixed effect model (Fixed factors: treatment 2 levels, year 2 levels) was used to assess the effects on conservation farming with and without biochar on SOC, total N, N mineralization rate and HWEC (Fig 2) over time at 0 to 8 cm and at 8

**Table 1. Grain yield and stover production in the growing seasons 2015–16, 2016–17, 2017–18 in Conservation farming (CF) with biochar (CF-BC), normal CF (CF-NORM), CF with no residue (CF-NO-RES) and conventional (CONV).** Standard error of the mean (s.e), n = 4. For root dimension n = 12.

| Growing season | | CF-BC | CF-NORM | CF-NO-RES | CONV |
|---|---|---|---|---|---|
| 2015–16 Maize | Grain (ton ha$^{-1}$) | 5.0[a] | 5.2[a] | 6.1[a] | 4.7[a] |
| | s.e | 0.4 | 0.6 | 0.4 | 0.4 |
| | Stover (ton ha$^{-1}$) | 4.9[a] | 4.4[a] | 5.3[a] | 3.7[a] |
| | s.e | 0.3 | 0.1 | 0.7 | 0.4 |
| 2016–17 Soya | Grain (ton ha$^{-1}$) | 3.0[a] | 3.4[a] | 3.6[a] | 2.1[b] |
| | s.e | 0.2 | 0.3 | 0.2 | 0.3 |
| | Stover (ton ha$^{-1}$) | 2.5[a] | 2.9[a] | 3.1[a] | 1.9[b] |
| | s.e | 0.3 | 0.3 | 0.2 | 0.2 |
| 2017–18 Maize | Grain (ton ha$^{-1}$) | 3.1[a] | 3.5[a] | 3.6[a] | 2.2[a] |
| | s.e | 0.2 | 0.3 | 0.3 | 0.5 |
| | Stover (ton ha$^{-1}$) | 5.7[a] | 4.9[a] | 5.7[a] | 4.7[a] |
| | s.e | 0.4 | 0.1 | 0.5 | 0.5 |
| | Root:shoot | 0.49[a] | 0.38[a] | - | 0.26[a] |
| | s.e | 0.21 | 0.10 | - | 0.04 |
| | *Root biomass (ton ha$^{-1}$; calculated) | 4.3[a] | 3.5[a] | - | 1.9[a] |
| | s.e | 1.9 | (1.0) | - | (0.5) |
| Root dimension | Max depth (cm) | 22.9 | 23.1 | - | - |
| | s.e | 0.6 | 0.4 | - | - |
| | Max width (cm) | 19.1 | 18.4 | - | - |
| | s.e | 1.1 | 0.8 | - | - |

*Root biomass was estimated only in 2017–18 in maize by multiplying root to shoot ratios and total biomass (grain + stover).

to 20 cm separately, where block was included as random factor. The spatial autocorrelation between repeated measurements in 2016 and 2018 was assumed constant between the different treatment combinations. Two-way ANOVA (treatment 2 levels and in vs. out 2 levels) was used to assess the difference in POM, POC, PON and C:N ratio of POM between CF-NORM and CF-NO-RES inside and outside basins in each depth independently. Differences between variables retained in the parsimonious models were analyzed by Tukey test at 0.05 significance. Linear regression was used to assess the relationship between HWEC and potential N mineralization rates.

# Results

## 3.1 Biomass production and grain yield

Soya yield and stover (season 2016–17), were significantly smaller under CONV (2.1 tons of grain ha$^{-1}$) than under CF-NORM, CF-NO-RES and CF-BC (3.4 tons, 3.6 tons and 3.0 tons of grain ha$^{-1}$, respectively; Table 1). Maize yield and maize stover production did not show any significant differences between treatments (i.e. seasons 2015–16 and 2017–18). Maize yields were higher in 2015–16 (5.2 ton grain ha$^{-1}$) than in the season 2017–18 (3.1 ton grain ha$^{-1}$). Retention of residue under CF-NORM corresponded to 4.4 ton ha$^{-1}$ maize stover in 2015–16 and 2.9 ton ha$^{-1}$ soya stover in 2016–17. Maize root to shoot ratios were not significantly different between treatments. The calculated root biomass was 3.2 ton ha$^{-1}$ in average (Table 1). The maximum depth and width of maize root system was 23 cm and 18.8 cm respectively (Table 1).

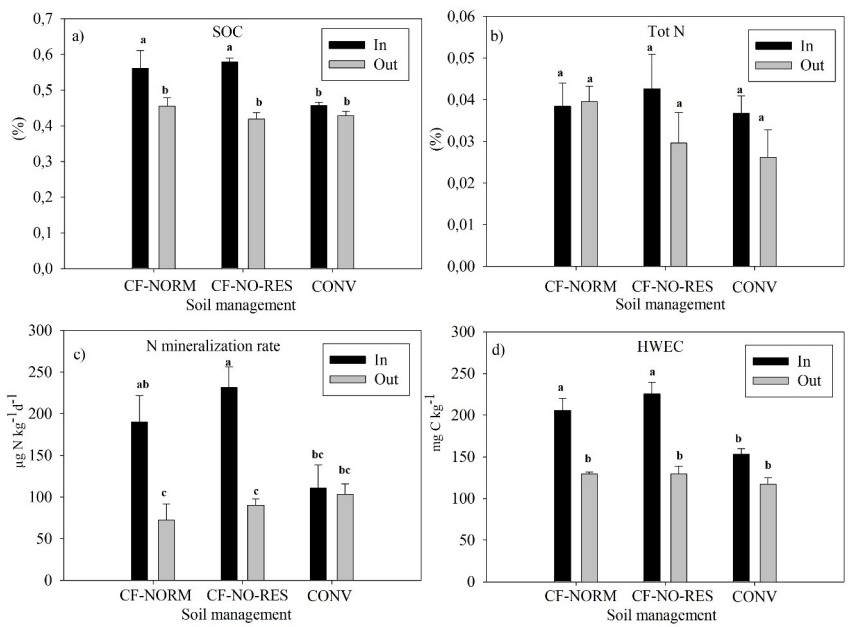

**Fig 1.** (a) Soil organic C, (b) total N, (c) N mineralization rate and (d) HWEC inside and outside basins under CF-NORM and CF-NO-RES and inside and outside planting rows in CONV in 2018 at 0 to 20 cm depth. The vales are depth weighted averages of the values from 0 to 8 cm and 8 to 20 cm. Error bars represent standard errors (n = 4). Lower case letters indicate significant differences (p<0.05) between soil management practices inside and outside basins (or inside and outside rows of plants under CONV).

### 3.2 Effect of conservation farming on SOC, SOC fractions, total N and N mineralization

There was no effect of residue removal on SOC, HWEC, total N and mineralizable N after 3 growing seasons (Fig 1, Table 2). Under CF-NORM and CF-NO-RES, SOC, HWEC and N mineralization rate were significantly greater inside basins as compared to outside (p<0.01 in both cases). Yet, total N did not significantly differ between inside and outside basins (Fig 1). Likewise, POM was not affected by residue removal (p = 0.51; Table 2). The results show larger amounts of POM inside basins than outside basins both at 0 to 8 cm and at 8 to 20cm (p<0.01 in all cases; Table 2). The quality of the POM was not affected by residue removal, as indicated by similar contribution of POC to SOC, PON to total N and CN ratio in POM in both treatments. Instead, the results show differences in the above-mentioned parameters between inside and outside basins (Table 2). In CONV, SOC, HWEC, total N and mineralizable N were the same inside and outside rows of plants. Total N did not show significant differences between the soil managements. Soil Organic C, HWEC and N mineralization rate were higher in CF systems inside basins than in CONV inside rows (Fig 1).

### 3.3 Effect of biochar, time and depth application inside basins

Biochar addition inside basins in CF significantly increased the amount of SOC as compared to CF-NORM at 8 to 20 cm depth where most of the biochar was added (Fig 2A). The amount of C added as biochar was 2.2 ton C ha$^{-1}$, while the difference in C stock inside basins in 2018 between CF-BC and either CF-NORM or CF-NO-RES corresponded to 2 ton C ha$^{-1}$ (Table 3), corresponding to a recovery of 90% of the biochar added. Total C stock per hectare in 2018 showed significant differences between CF-BC and CONV only (p<0.01; Table 3). However, CF-BC had significantly higher C stock inside basins than both CF-NORM and CF-NO-RES

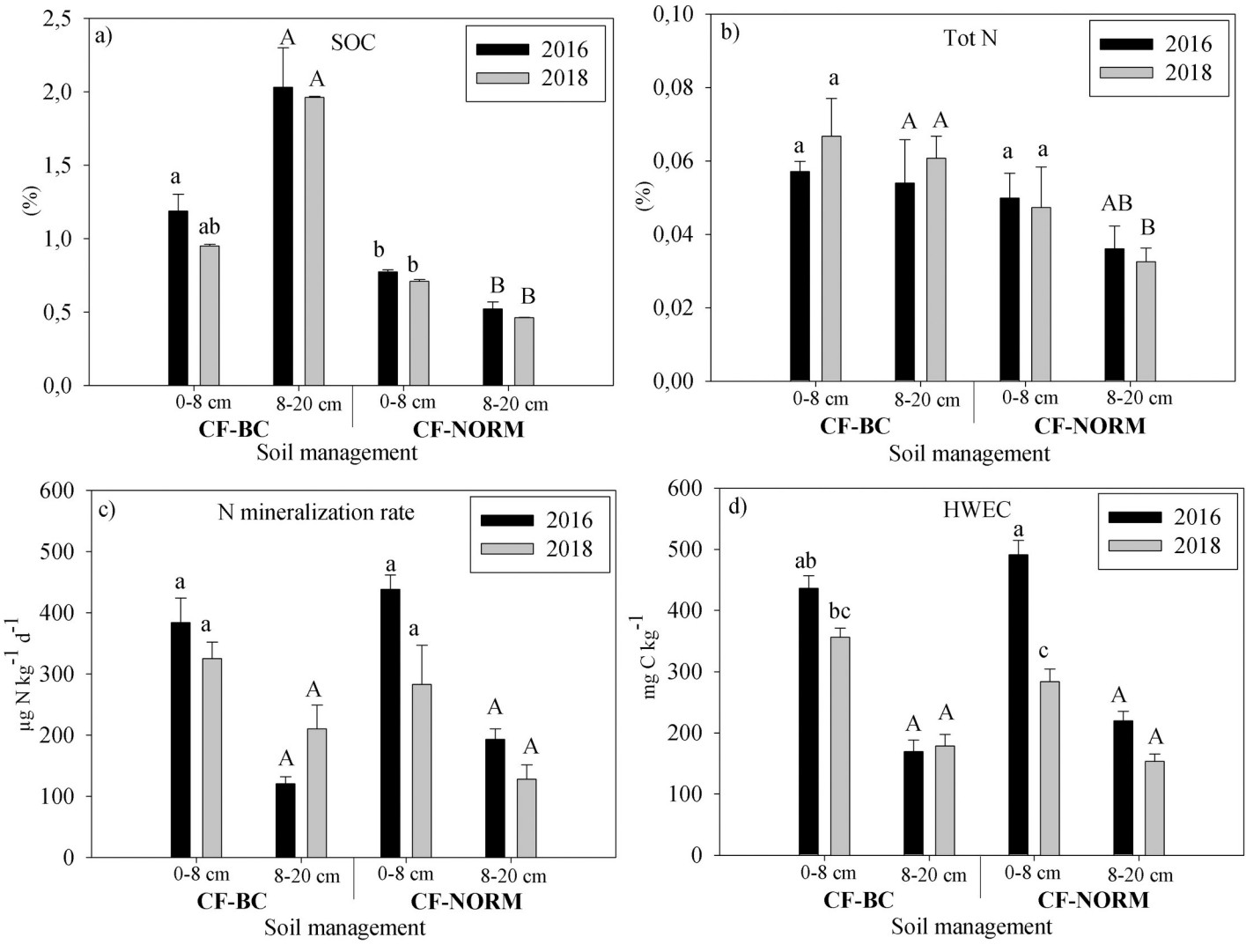

**Fig 2.** (a) SOC, (b) total N, (c) N mineralization rate and (d) HWEC inside planting basins in CF-NORM and CF-BC at 0 to 8 cm and 8 to 20 cm depth in 2016 and 2018. Lower case letters denote differences between treatments in 2016 and 2018 at 0 to 8 cm. Upper case letters show differences between treatments in 2016 and 2018 at 8 to 20 cm. Error bars represent standard errors, n = 4.

(p < 0.01 in both). In 2018, concentration of total N (%) was increased upon biochar addition as compared to CF-NORM at 8 to 20 cm only (Fig 2B). However, results showed no differences in N stock per hectare when CF-NO-RES, CF-NORM and CF-BC were included in the analysis (Table 3). Nitrogen mineralization rates and HWEC under CF-BC did not significantly differ from CF-NORM in 2016 and 2018 (Fig 2). Biochar increased amount of SOC but not its more labile fraction, as estimated by hot water extraction (HWEC). The correlation of N mineralization rate and HWEC was clear and it was not affected by any of the soil regimes including CF-BC ($R^2$ = 0.81; Fig 3).

## Discussion

### 4.1 Crop biomass-effect of soil management

Soya yield and stover were significantly lower in CONV than in CF systems in 2016–17 (Table 1), when no fertilizer was applied. For maize, we found no significant effect of treatment

**Table 2. Particulate organic matter in the bulk soil, contribution of particulate organic C (POC) to SOC, contribution of particulate organic N (PON) to total N and CN ratio in POM in CF-NORM and CF-NO-RES inside and outside basins at 0 to 8 cm and 8 to 20 cm in 2018.** Lowercase letters denote differences in POM and quality of POM between inside and outside basins independent of the soil management treatment. Values are averages with standard errors, n = 4.

| | POM in bulk soil (%) | | POC to SOC (%) | | PON to Tot N (%) | | C:N in POM | | C:N in bulk soil | |
|---|---|---|---|---|---|---|---|---|---|---|
| | CF-NO-RES | CF-NORM | CF-NO-RES | CF-NORM | CF-NO-RES | CF-NORM | CF-NO-RES | CF-NORM | CF-NO-RES | CF-NORM |
| Inside | [a] | | [a] | | [a] | | [b] | | [a] | |
| 0–8 cm | 0.82± 0.02 | 0.81± 0.13 | 30.8±0.8 | 33.2± 3.9 | 31.8±11.2 | 33.9± 10.4 | 16.0± 0.6 | 17.3± 0.5 | 15.8± 4.6 | 17.0± 3.4 |
| 8–20 cm | 0.43± 0.03 | 0.44± 0.06 | 21.5±1.2 | 21.9± 2.3 | 17.6±5.2 | 15.0± 2.2 | 20.5± 0.4 | 21.4± 1.7 | 17.2± 5.6 | 14.5± 1.4 |
| Outside | [b] | | [b] | | [b] | | [a] | | [a] | |
| 0–8 cm | 0.40± 0.03 | 0.44± 0.03 | 23.7±0.7 | 22.8± 2.7 | 19.6±3.7 | 13.2± 2.7 | 20.7± 0.8 | 19.7± 0.7 | 17.3± 3.7 | 11.3± 1.4 |
| 8–20 cm | 0.20± 0.02 | 0.26± 0.02 | 13.9±0.5 | 18.3± 1.6 | 10.2±5.1 | 6.0± 2.0 | 30.8± 6.4 | 45.8± 5.7 | 18.9± 7.0 | 13.5± 2.6 |

(including biochar) on yield. Possibly, the differences in spatial planting patterns within maize rows in CF-NORM vs CONV explained the lack of significant differences in maize biomass, despite the benefits on soil fertility of CF-NORM. As reported by Mashingaidze et al. [39], maize plants planted at short distances as inside basins may be hampered by competition for nutrients, water and radiation compared with plants in CONV, which are more widely spaced. Soya in CONV was sown in planting stations that had the same spatial arrangement as CF-NORM.

Maize yields in our experiment were greater (Table 1) than those generally obtained by small-scale farmers in SSA, which were estimated to be 1.4 ± 1.0 ton ha$^{-1}$ for maize [40]. The reason is likely to be the high N application rate (116 kg N ha$^{-1}$) in our experiment. By contrast, small-scale farmers in SSA use on average 17 kg NPK ha$^{-1}$ [14]. Consequently, amounts of maize and soya residues were high (4.4 and 2.9 ton ha$^{-1}$ y$^{-1}$, respectively). Research conducted in SSA has suggested 3 ton ha$^{-1}$ of crop residues as threshold value to reach an initial soil cover of 30% of the land at the beginning of each growing season [41]. Often, small-scale farmers do not manage to produce that amount of residues. The reason for higher maize yield in 2015–16 than in 2017–18 was likely due to a dry spell that affected Zambia mainly during January 2018 (S3 Fig).

Results showed no effect of soil treatment (including biochar addition) on root to shoots ratio. Previously, Abiven et al. Abiven, Hund [42] found more developed maize roots systems and greater yields in biochar amended plots compared to normal CF plots on the same site in

**Table 3. Carbon and Nitrogen stocks per hectare in the upper 20 cm in CF-NORM, CF-NO-RES, CONV and CF-BC in 2018.** C and N stocks inside basins were compared between Conservation Farming (CF) systems only. In the three CF treatments, area dedicated to planting basins was 9.7% of the field whereas outside basins was 90.3%. The comparison of total C and N stock included CF systems as well as CONV. C and N stocks outside basins in CF-BC were assumed the same as in CF-NORM. Lower case letters indicate significant differences in C and N stocks between treatments, comparing either inside, outside basins or the total C stock per hectare (n = 4).

| | | Soil management practices | | | |
|---|---|---|---|---|---|
| | | CF-BC | CF-NORM | CF-NO-RES | CONV |
| C stock | Inside | 3.5 (0.32)[a] | 1.5 (0.13)[b] | 1.5 (0.03)[b] | - |
| (ton C ha$^{-1}$) | Outside | | 11.2 (0.60)[a] | 11.0 (0.48)[a] | - |
| | Total | 14.1 (0.3)[a] | 12.7 (0.6)[ab] | 12.5 (0.5)[ab] | 11.4 (0.3)[b] |
| N stock | Inside | 143 (17)[a] | 100 (14)[a] | 111 (21)[a] | - |
| (kg N ha$^{-1}$) | Outside | | 976 (92)[a] | 779 (192)[a] | - |
| | Total | 1029(13.3)[a] | 1076 (93)[a] | 890 (205)[a] | 739 (159)[a] |

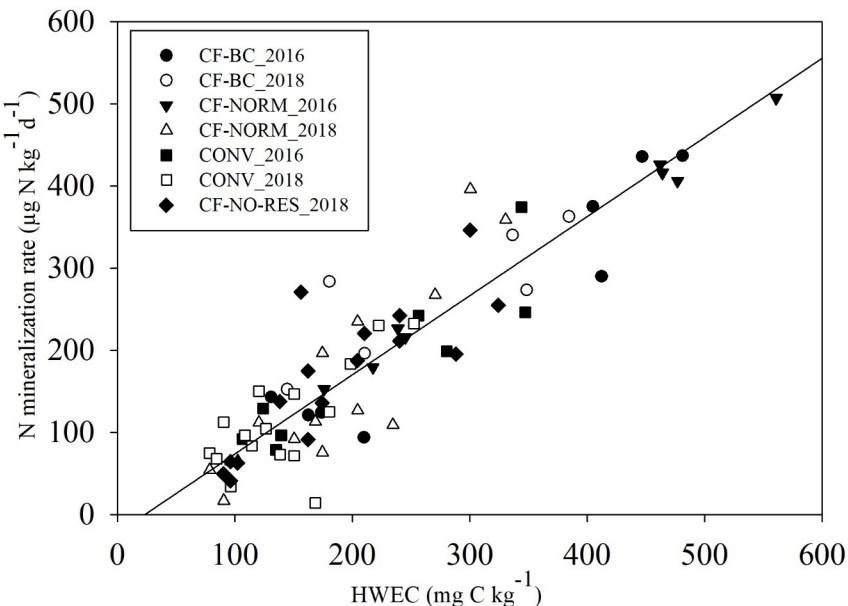

**Fig 3. N mineralization rates and hot-water extractable C.** Relationship between HWEC and potential N mineralization rates in CF-BC, CF-NORM, CF-NO-RES and CONV in the sampling campaigns of 2016 and 2018 [$R^2$ = 0.82; N min rate ($\mu$g-N kg soil$^{-1}$d$^{-1}$) ~ 0.9622 x HWEC (mg C kg soil$^{-1}$)– 22.359].

Zambia. However, the application of N in their Abiven, Hund [42] was only 30 kg N ha$^{-1}$ and the root to shoot ratio was 0.037 to 0.045, which is much smaller than found in our study (0.26 to 0.49, Table 1). Our values are in the range of what was reported for maize in North America [43, 44]. Results showed estimates of maize root biomass that varied from 1.9 ton ha$^{-1}$ in CONV to 3.5 ton ha$^{-1}$ in CF-BC (Table 1).

## 4.2 Effect of permanent planting basins, tillage and residue retention on SOC, C fractions, total N and N mineralization

The experimental field at Mkushi had been managed with permanent planting basins and residue retention between rows of basins for 7 years prior the establishment of the treatments. At the onset of the present study SOC, HWEC and N mineralization rate were therefore higher inside permanent planting basins as compared to outside [33]. Upon full tillage in CONV management, SOC, N mineralization rate and HWEC decreased to the same levels observed for the area outside basins in CF (Fig 1). This could be explained by the redistribution of soil upon conventional tillage that diluted the HWEC that was gained inside basins. Other studies have shown sharp long-term decrease of HWEC and other labile SOC fractions upon full tillage has been reported in other tropical, subtropical as well as temperate soils mainly in the upper 5 cm [24, 45]. HWEC and N mineralization were highly correlated (Fig 3), therefore, N mineralization in CONV was also lower than in CF inside basins (Fig 1). This suggests that it takes few seasons of conventional tillage to undo the long-term benefits of conservation faming on soil C and N mineralization.

Residue removal is the common practice in Sub-Saharan Africa since there are competing uses such as animal feed or fuel [46].The results suggests no effect of residue removal after nearly three growing seasons on amount and quality of SOM (Fig 1, Table 2). Previous research has suggested limited effect of crop residues on SOC increments in soils in South Africa and Kenya in experiments conducted over longer periods of time than in the present

experiment [47, 48]. This may be due to high decomposition rates of crop residues [49]. However, residue losses due to termite activity, which increases with increasing residues [50], or transport by wind and water may also have contributed to the lack of effect of residues on SOC and its particulate and soluble fractions. The fact that residues do not have an effect on either SOM or POM ($<$ 2 mm) does not imply that they do not provide other benefits such as soil moisture retention and weed control if they are kept as mulch [51].

Amount of SOC, HWEC and N mineralization inside basins were greater than outside (Fig 1; Table 2). The accumulation of root-derived biomass inside basins may explain this pattern as suggested by Martinsen et al. [33]. Inside basins (10% of the area), about 3.5 ton ha$^{-1}$ of root biomass in CF-NORM (Table 1) was concentrated and incorporated into the soil, while outside basins about 4.5 ton ha$^{-1}$ (in the case of maize) of crop residues was distributed over a 10 times larger area and placed on soil surface. Previous research has shown that root biomass is retained as SOM more efficiently than shoot biomass [52]. Often this has been attributed to smaller decomposability of roots compared to shoots [44, 52]. The observation that POM was higher inside basin than outside at both 0 to 8 am and 8 to 20 cm also favors the idea of root carbon retention, since crop residue biomass is applied at the surface while root biomass can penetrate deeper (Table 2).

Our results indicate no further increase of SOC in conservation farming inside basins after seven years Martinsen, Munera-Echeverri (33). This may suggest that more time is needed to detect significant increases in SOC. Also, the rotation maize-groundnuts (before the 7$^{th}$ year) could be more effective increasing SOC than maize-soya due to a higher root biomass derived from the larger N fixation capacity of groundnuts vs soya In a rotation with rice, groundnut was found to fix more N than soybean (150 to 200 kg N ha$^{-1}$ vs 108 to 150 kg N ha$^{-1}$, respectively) and to increase rice biomass and grain yield [53]. An important amount of N (from 100 to 130 kg N ha$^{-1}$) has been measured in groundnut stover [54], which may increase maize yield if returned to soil [54, 55], as it is commonly done in Zambia.

## 4.3 Effect of biochar on amount and quality of SOC over time and depth

CF-BC significantly increased SOC and C stocks inside basins as compared to CF-NORM and CF-NO-RES (Fig 2; Table 3). In addition, C stock per hectare in CF-BC was significantly higher than in CONV (Table 3). The reason for not finding significant differences between in C stocks per hectare between CF-BC and CF-NORM may be attributed to the fact that biochar was added only to about 10% of the land and most of the C in the field was found outside basins (Table 3). C stocks in the present study were low as compared to the values reported by Martinsen et al. [56] for other soils in the Eastern and Central provinces in Zambia in conventional and CF fields (28 to 45 ton C ha$^{-1}$ vs. about 12.7 ton C ha$^{-1}$). The high recovery of biochar (90%) 2.5 years after addition suggests high stability of biochar in agreement with Kuzyakov et al. Kuzyakov, Bogomolova [57], as well as limited lateral BC transport, i.e. floating followed by erosion. Lateral BC transport was observed to be up to 30–40% in a similar soil in Zambia [34]. Vertical BC transport was observed to be limited to 1–2 cm per year, and thus vertical transport to below 20 cm depth would not be expected. Despite larger SOC in CF-BC, HWEC was unaffected by biochar addition (Fig 2). HWEC decreased with depth with and without biochar. HWEC has been found to decrease with depth in uncultivated, agricultural and forest soils [58, 59]. Likewise, CF-BC did not affect N mineralization rate and consequently, we did not find evidence for positive or negative priming effect of biochar on SOM decomposition.

The net N mineralization rate increased with about 96 µg-N kg$^{-1}$d$^{-1}$ when HWEC increased by 100 mg kg$^{-1}$ (Fig 3). HWEC was about 2.3% of the total SOC in CF-BC, 4.0% in CF-NORM,

and 3.4% in CONV. The decrease of HWEC from 2016 to 2018 in CF-NORM (Fig 2) may be partially explained by the differences in maize root C inputs throughout the growing season, since sampling in 2016 was done at the end of the season (maximum amount of C inputs) whereas in 2018 it was done in the middle. In addition, maize biomass in the season 2017–18 decreased as compared to 2015–16, probably due to water-stress. Fluctuations in HWEC across seasons has been reported in pine forest in Korea and this was linked to changes in the amount of extractable carbohydrates [59]. Soluble carbohydrates have been found to constitute about 79% of the root exudates of maize plants [60]. In addition, soil pH inside basins decreased during the experiment from 6.3±0.2 [33] to 4.5 (with and without biochar) due to no continuation of liming. This could also explain the decrease in HWEC since solubility of SOM increases at increasing pH [61]. The decrease in HWEC from 2016 to 2018 can explain the decrease in N mineralization rate since DOM is an important substrate for microbial activity [62].

## Conclusions

Soya shows increase yields under conservation farming as compared to conventional farming. Maize yield does not show response to conservation or conventional farming, due to the difference in planting arrangements and/or the addition of the ample addition of fertilizer when maize was planted. Soil organic carbon, particulate organic matter, and hot-water extractable carbon were larger inside basins than outside basins in conservation farming due to the continuous allocation of root biomass inside basins. Nitrogen mineralization was enhanced inside basins due to the increase of labile C. After conventional tillage of conservation farming plots, soil organic carbon, hot-water extractable carbon, and mineralizable nitrogen inside basins decrease. Residue removal does not have a significant effect on crop yield, soil organic carbon, hot-water extractable carbon, total nitrogen, nitrogen mineralization rates, and particulate organic matter, which agrees with previous studies. The addition of pigeon pea biochar to planting basins under conservation farming is effective in increasing amounts of soil organic carbon as compared to either the normal practice of conservation farming or the conventional practice. However, biochar does not affect hot-water extractable carbon, nitrogen mineralization rate, and crop biomass.

## Supporting information

**S1 Fig.** Soil regimes: a) CF-NORM: residue retention, permanent basins, b) CF-NO-RES: no crop residues without residue retention, c) CF-BC: addition of pigeon pea biochar inside basins and d) CONV: full tillage at a depth of 20 cm and no residue retention.
(TIF)

**S2 Fig.** a) Experimental setup. The four soil regimes were randomly distributed in 4 blocks. Each plot was about 20 m2 and consisted of 4 rows of six basins in all the Conservation Farming (CF) treatments and 4 rows of plants in conventional (CONV) plots. b) CF plots. Each row consisted of 6 planting basins with 3 plants of maize. c) CONV plots. Each row had 18 plants.
(TIF)

**S3 Fig. Precipitation from October 2017 to October 2018 recorded in the weather station of the experimental field.** Precipitation from November 2017 to January 2018 was unusually low. The effect of the dry-spell on Zambia's maize harvest was mentioned in the press: https://www.bloomberg.com/news/articles/2018-05-04/dry-spell-slashes-zambian-corn-production-by-34-in-2017-18
(TIFF)

**S1 File. Excel file with the results of the density fractionation of soil organic matter.** In the sheet ALL_DATA_WP. The columns "Amount POM (percent)", "C_in POM", "N_in POM" and "CN_POM" correspond to the values shown in Table 2.
(XLSX)

**S2 File. Excel file with the biomass data.** The soya biomass is found in the sheet "ORG-Soya" and maize biomass in 2016 and 2018 is found in the sheet "ORG-Maize".
(XLSX)

**S3 File. Excel file with the carbon and nitrogen stocks per hectare inside and outside basins in CF-BC, CF-NORM, CF-NO-RES and CONV.**
(XLSX)

**S4 File. Soil carbon and nitrogen data collected in 2016 and 2018 under CF-BC, CF-NORM, CF-NO-RES and CONV.** The data include nitrogen mineralization rates after 60 days of incubation and hot-water extractable C.
(XLSX)

**S1 Table. Soil pH inside and outside basins in 2016 and 2018.**
(DOCX)

## Acknowledgments

The study was funded by the Faculty of Environmental Sciences and Natural Resource Management at the Norwegian University of Life Sciences and by the R&D project "Soil Benefits Parameters Research, Conventional and Conservation Agriculture" funded by the Conservation Farming Unit (CFU) in Zambia with support from DFID. We are grateful to Victor Shitumbanuma for facilitating the permission for transportation of the soil samples. Thanks to Jeremy Selby for allowing access to his fields, for careful management of the research plots and for his hospitality. Thanks to Edward Bwalya and Kelvin for their help during the field work.

## Author Contributions

**Conceptualization:** Jose Luis Munera-Echeverri, Vegard Martinsen, Line Tau Strand, Gerard Cornelissen, Jan Mulder.

**Data curation:** Jose Luis Munera-Echeverri, Vegard Martinsen.

**Formal analysis:** Jose Luis Munera-Echeverri, Vegard Martinsen.

**Funding acquisition:** Vegard Martinsen, Gerard Cornelissen, Jan Mulder.

**Investigation:** Jose Luis Munera-Echeverri, Jan Mulder.

**Methodology:** Jose Luis Munera-Echeverri, Vegard Martinsen, Line Tau Strand, Gerard Cornelissen, Jan Mulder.

**Project administration:** Vegard Martinsen, Jan Mulder.

**Resources:** Vegard Martinsen, Gerard Cornelissen, Jan Mulder.

**Supervision:** Vegard Martinsen, Line Tau Strand, Gerard Cornelissen, Jan Mulder.

**Validation:** Jose Luis Munera-Echeverri, Vegard Martinsen, Line Tau Strand.

**Visualization:** Line Tau Strand.

**Writing – original draft:** Jose Luis Munera-Echeverri.

**Writing – review & editing:** Jose Luis Munera-Echeverri, Vegard Martinsen, Line Tau Strand, Gerard Cornelissen, Jan Mulder.

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
