## [Decision Letter · Decision Letter 0]

7 Nov 2019

PONE-D-19-27232

Effect of conservation farming and biochar addition on SOC quality, N mineralization, and crop productivity in a light textured Acrisol in the sub-humid tropics

PLOS ONE

Dear Dr Munera-Echeverri,

Thank you for submitting your manuscript to PLOS ONE. After careful consideration, we feel that it has merit but does not fully meet PLOS ONE’s publication criteria as it currently stands. Therefore, we invite you to submit a revised version of the manuscript that addresses the points raised during the review process.

Please, address all comments by both reviewers.

We would appreciate receiving your revised manuscript by Dec 22 2019 11:59PM. To enhance the reproducibility of your results, we recommend that if applicable you deposit your laboratory protocols in protocols.io, where a protocol can be assigned its own identifier (DOI) such that it can be cited independently in the future. For instructions see: http://journals.plos.org/plosone/s/submission-guidelines#loc-laboratory-protocols

We look forward to receiving your revised manuscript.

Kind regards,

Jorge Paz-Ferreiro, Ph.D.

Academic Editor

PLOS ONE

Journal Requirements:

Reviewers' comments:

Reviewer's Responses to Questions

**Comments to the Author**

1. Is the manuscript technically sound, and do the data support the conclusions?

Reviewer #1: Partly

Reviewer #2: Yes

2. Has the statistical analysis been performed appropriately and rigorously? 

Reviewer #1: Yes

Reviewer #2: Yes

3. Have the authors made all data underlying the findings in their manuscript fully available?

Reviewer #1: Yes

Reviewer #2: Yes

4. Is the manuscript presented in an intelligible fashion and written in standard English?

Reviewer #1: Yes

Reviewer #2: Yes

5. Review Comments to the Author

Reviewer #1: - Please include a more robust description of the experimental design. Specifically, focus on replications, sample sizes for each of the dependent variables. I had to spend time looking through supplementary materials to find this information.

- The planting arrangements in the maize varies and needs to be discussed as another potential contributing factor to the lack of yield differences in your trial.

- You might spend more time discussing the importance of the maize-groundnut system and why that was resulted in differences compared to your study.

- I would also encourage you to add more discussion regarding the 7 years prior in CF as this also likely influenced the results of your study.

Reviewer #2: Dear Authors,

I am enclosing herewith the review comments of the manuscript entitled "Effect of conservation farming and biochar addition on SOC quality, N mineralization, and crop productivity in a light textured Acrisol in the sub-humid tropics". The study presented by you is a nice representing of work on conservation farming practices along with biochar application and its effect on soil health aspects (soil organic matter, N-mineralization, etc,). This piece of work is very relevant to the present context of soil health and as a means of environmental conservation practices. Many papers are available on biochar but a combination of biochar and conservational farming, tillage, crop rotation on a long-term basis is still needed for the scientific community. Therefore, this paper would be a relevant and good resource for other researchers throughout the world. However, I do have some moderate revisions/suggestions/recommendations. I strongly recommend to go through the comments, make changes or refute with proper explanations as needed to get finally accepted to our journal. I have very general and some specific comments for this paper. I would only consider this manuscript only when these following MODERATE changes (MODERATE REVISION; I’m giving major because there is no moderate revision option for our journal) will be made or suggestions are refuted with proper explanation. My comments are intended to further improve the manuscript such that it addresses the questions as asked properly. I would reconsider the manuscript for publication if the moderate/minor major changes are made and resubmitted.

General comments

1. Suggest not to use any abbreviation in the title of the manuscript.

2. Recommend to use ‘Table’ not ‘tab’ in the entire manuscript. Please make change accordingly.

3. Materials and methods are very detailed-oriented and explanatory which is good for readers.

Specific comments

Introduction

1. In general, authors have used good reference to refer their manuscript. However, I recommend and suggest to read some articles quickly which are useful, recent and very relevant articles to cite in this manuscript not only to make the literature search up-to-date but also the paper will be more acknowledged while readers’ read it. Please see my line-wise comments.

2. Line 41-44: Please use some recent references in some reputed journal which will explain many recent facts in this manuscript such as Sihi et al., 2016, Sihi et al., 2017 (see the full reference list below). You can use those papers and may want to cite it wherever you think is relevant.

3. Line 47-49: Please use these recently published articles whichever you think is relevant: Chalise et al., 2019, Maiga et al., 2019, Sihi et al., 2017.

4. Line 54-56: You may want to include the citation from this perspective paper and research papers which are recent and very related: Nair et al., 2017 and Dari et al., 2016.

5. Line 68-69: A relevant paper area on conservation farming practices on soil organic matter and soil health in tropical/sub-tropics areas can be cited here: Sihi et al., 2017.

6. Line 82-85: Please read these papers and cite accordingly: Chalise et al., 2019, Maiga et al., 2019.

7. Line 93-98: “This study shows positive effects…climate-smart solutions for agriculture in sub-humid regions”…….this whole section should not be the part of ‘Introduction’. It should be part of your result section. Please move to the results and Discussion section or rather conclusion section.

References

1. Sihi, D., Gerber, S., Inglett, P. W., & Inglett, K. S. (2016). Comparing models of microbial–substrate interactions and their response to warming. Biogeosciences, 13(6), 1733-1752.

2. Sihi, D., Inglett, P. W., Gerber, S., & Inglett, K. S. (2018). Rate of warming affects temperature sensitivity of anaerobic peat decomposition and greenhouse gas production. Global change biology, 24(1), e259-e274.

3. Chalise, K.S., Singh, S., Wegner, B.R., Kumar, S., Pérez-Gutiérrez, J.D., Osborne, S.L., Nleya, T., Guzman, J., Rohila, J.S., 2019. Cover Crops and Returning Residue Impact on Soil Organic Carbon, Bulk Density, Penetration Resistance, Water Retention, Infiltration, and Soybean Yield. Agronomy Journal 111, 99-108.

4. Maiga, A., Alhameid, A., Singh, S., Polat, A., Singh, J., Kumar, S., Osborne, S., 2019. Responses of soil organic carbon, aggregate stability, carbon and nitrogen fractions to 15 and 24 years of no-till diversified crop rotations. Soil Research.

5. Sihi, D., Dari, B., Sharma, D. K., Pathak, H., Nain, L., & Sharma, O. P. (2017). Evaluation of soil health in organic vs. conventional farming of basmati rice in North India. Journal of Plant Nutrition and Soil Science, 180(3), 389-406.

6. Nair, V. D., Nair, P. K., Dari, B., Freitas, A. M., Chatterjee, N., & Pinheiro, F. M. (2017). Biochar in the agroecosystem–climate-change–sustainability nexus. Frontiers in plant science, 8, 2051.

7. Dari, B., Nair, V. D., Harris, W. G., Nair, P. K. R., Sollenberger, L., & Mylavarapu, R. (2016). Relative influence of soil-vs. biochar properties on soil phosphorus retention. Geoderma, 280, 82-87.

Materials and Methods

1. Line 113-114: Why biochar was applied in that particular depth, why not other depth?

2. Rest of the experimental details and all section in M&M part is well explained and nicely described. I do to have any comments.

Tables

1. Make sure the title of each tables are not bold. Be consistent and there is no need to bold any words within the table even.

2. Same comments as above for figures (no bold except the word TABLE or FIG.), make the relevant changes.

3. I do not see any reason to highlight some columns in your table (e.g. Table 2) so remove it.

Conclusions

1. Strongly recommend to make your conclusion succinct. Please re-write it having only those part which are direct results of your study.

2. Please do not include any reference or citation or others’ results in the conclusion section e.g. remove the line 387-389 from the conclusion section to your introduction part or Discussion part.

3. Suggest not to have multiple paragraphs for conclusion. Try to have one solid paragraphs with all important conclusion of your study unless it is really necessary to stretch your main results.

6. PLOS authors have the option to publish the peer review history of their article (what does this mean?). If published, this will include your full peer review and any attached files.

Reviewer #1: No

Reviewer #2: No

---

## [Author Response · Author response to Decision Letter 0]

6 Dec 2019

December 4, 2019

Dear dr. Paz-Ferreiro,

We are pleased to submit the revised version of our manuscript “Effect of conservation farming and biochar addition on soil organic carbon quality, nitrogen mineralization, and crop productivity in a light textured Acrisol in the sub-humid tropics”. Added text in the revised version is highlighted in green and removed text with track changes. The Ethics Statement has been updated in the submission and it has been included in the Materials and methods section in lines 96-101.

Thanks for the quick review of our manuscript. We thank the reviewers for their insightful comments

5. Review Comments to the Author

Reviewer #1: - Please include a more robust description of the experimental design. Specifically, focus on replications, sample sizes for each of the dependent variables. I had to spend time looking through supplementary materials to find this information.

Thanks for the constructive recommendation. We have now explicitly described the details in the Biomass production (lines 140-141) and Soil sampling (line 149 and lines 159-162) s sections of materials and methods. In addition, in the legend of each figure and table we included the number of replicates. 

- The planting arrangements in the maize varies and needs to be discussed as another potential contributing factor to the lack of yield differences in your trial.

Done. We discuss this point in lines 301-308 and we also included a discussion of the spatial arrangement of soya. We improved the description of the arrangement of both maize and soya in the Materials and Methods in lines 119-122.

- You might spend more time discussing the importance of the maize-groundnut system and why that was resulted in differences compared to your study.

Done. In the revised version in lines 364-371, we discuss the importance of the maize-groundnut system and provided three new references. 

- I would also encourage you to add more discussion regarding the 7 years prior in CF as this also likely influenced the results of your study.

We reorganized the Discussion and the Conclusions sections, in agreement with the comments of the second reviewer. We include the discussion about the 7 years in Line 328-338 and in lines 364-371. We discuss that the selected soil parameters were higher inside basins than outside basins after 7 years of CF with maize-groundnut rotation, and that from 7 to 9.5 years, i) there were no measurable increments of soil organic matter in CF-NORM and that ii) upon tillage, the gains of SOC inside basins were lost.

Reviewer #2: Dear Authors,

I am enclosing herewith the review comments of the manuscript entitled "Effect of conservation farming and biochar addition on SOC quality, N mineralization, and crop productivity in a light textured Acrisol in the sub-humid tropics". The study presented by you is a nice representing of work on conservation farming practices along with biochar application and its effect on soil health aspects (soil organic matter, N-mineralization, etc,). This piece of work is very relevant to the present context of soil health and as a means of environmental conservation practices. Many papers are available on biochar but a combination of biochar and conservational farming, tillage, crop rotation on a long-term basis is still needed for the scientific community. Therefore, this paper would be a relevant and good resource for other researchers throughout the world. However, I do have some moderate revisions/suggestions/recommendations. I strongly recommend to go through the comments, make changes or refute with proper explanations as needed to get finally accepted to our journal. I have very general and some specific comments for this paper. I would only consider this manuscript only when these following MODERATE changes (MODERATE REVISION; I’m giving major because there is no moderate revision option for our journal) will be made or suggestions are refuted with proper explanation. My comments are intended to further improve the manuscript such that it addresses the questions as asked properly. I would reconsider the manuscript for publication if the moderate/minor major changes are made and resubmitted.

We thank the reviewer for the constructive feedback and useful suggestions. 

General comments

1. Suggest not to use any abbreviation in the title of the manuscript.

Done.

2. Recommend to use ‘Table’ not ‘tab’ in the entire manuscript. Please make change accordingly.

Done.

3. Materials and methods are very detailed-oriented and explanatory which is good for readers.

Thanks for the positive comments.

Specific comments

Introduction

1. In general, authors have used good reference to refer their manuscript. However, I recommend and suggest to read some articles quickly which are useful, recent and very relevant articles to cite in this manuscript not only to make the literature search up-to-date but also the paper will be more acknowledged while readers’ read it. Please see my line-wise comments.

2. Line 41-44: Please use some recent references in some reputed journal which will explain many recent facts in this manuscript such as Sihi et al., 2016, Sihi et al., 2017 (see the full reference list below). You can use those papers and may want to cite it wherever you think is relevant.

3. Line 47-49: Please use these recently published articles whichever you think is relevant: Chalise et al., 2019, Maiga et al., 2019, Sihi et al., 2017.

Done

4. Line 54-56: You may want to include the citation from this perspective paper and research papers which are recent and very related: Nair et al., 2017 and Dari et al., 2016.

Done

5. Line 68-69: A relevant paper area on conservation farming practices on soil organic matter and soil health in tropical/sub-tropics areas can be cited here: Sihi et al., 2017.

We did not include this paper since it does not refer to particulate organic matter, which is the topic we discuss in these lines.

6. Line 82-85: Please read these papers and cite accordingly: Chalise et al., 2019, Maiga et al., 2019.

These two papers were already cited. However, we think they do not fit in these lines since we discuss the specific case of the practices followed by smallholder farmers in Sub-Saharan Africa. 

7. Line 93-98: “This study shows positive effects…climate-smart solutions for agriculture in sub-humid regions”…….this whole section should not be the part of ‘Introduction’. It should be part of your result section. Please move to the results and Discussion section or rather conclusion section.

References

1. Sihi, D., Gerber, S., Inglett, P. W., & Inglett, K. S. (2016). Comparing models of microbial–substrate interactions and their response to warming. Biogeosciences, 13(6), 1733-1752.

2. Sihi, D., Inglett, P. W., Gerber, S., & Inglett, K. S. (2018). Rate of warming affects temperature sensitivity of anaerobic peat decomposition and greenhouse gas production. Global change biology, 24(1), e259-e274.

3. Chalise, K.S., Singh, S., Wegner, B.R., Kumar, S., Pérez-Gutiérrez, J.D., Osborne, S.L., Nleya, T., Guzman, J., Rohila, J.S., 2019. Cover Crops and Returning Residue Impact on Soil Organic Carbon, Bulk Density, Penetration Resistance, Water Retention, Infiltration, and Soybean Yield. Agronomy Journal 111, 99-108.

4. Maiga, A., Alhameid, A., Singh, S., Polat, A., Singh, J., Kumar, S., Osborne, S., 2019. Responses of soil organic carbon, aggregate stability, carbon and nitrogen fractions to 15 and 24 years of no-till diversified crop rotations. Soil Research.

5. Sihi, D., Dari, B., Sharma, D. K., Pathak, H., Nain, L., & Sharma, O. P. (2017). Evaluation of soil health in organic vs. conventional farming of basmati rice in North India. Journal of Plant Nutrition and Soil Science, 180(3), 389-406.

6. Nair, V. D., Nair, P. K., Dari, B., Freitas, A. M., Chatterjee, N., & Pinheiro, F. M. (2017). Biochar in the agroecosystem–climate-change–sustainability nexus. Frontiers in plant science, 8, 2051.

7. Dari, B., Nair, V. D., Harris, W. G., Nair, P. K. R., Sollenberger, L., & Mylavarapu, R. (2016). Relative influence of soil-vs. biochar properties on soil phosphorus retention. Geoderma, 280, 82-87.

Materials and Methods

1. Line 113-114: Why biochar was applied in that particular depth, why not other depth?

Done. It can be found in lines 113-116 of the revised version.

“The planting basins were dug, using a hoe, to a depth of 20 cm, in agreement with the local practice of conservation farming; and biochar was added at a depth of 20 cm, mixed with the soil and subsequently, covered with more soil. Most of the biochar was placed at a depth of 8 to 20 cm.”

2. Rest of the experimental details and all section in M&M part is well explained and nicely described. I do to have any comments.

Thanks.

Tables

1. Make sure the title of each tables are not bold. Be consistent and there is no need to bold any words within the table even.

Done.

2. Same comments as above for figures (no bold except the word TABLE or FIG.), make the relevant changes.

Done.

3. I do not see any reason to highlight some columns in your table (e.g. Table 2) so remove it.

Done.

Conclusions

Thanks for the comments of this section. In the revised version, we extended the Discussion, made the Conclusions succinct and now it is a single paragraph.

1. Strongly recommend to make your conclusion succinct. Please re-write it having only those part which are direct results of your study.

Done. 

2. Please do not include any reference or citation or others’ results in the conclusion section e.g. remove the line 387-389 from the conclusion section to your introduction part or Discussion part.

Done.

3. Suggest not to have multiple paragraphs for conclusion. Try to have one solid paragraphs with all important conclusion of your study unless it is really necessary to stretch your main results.

Done.

---

## [Decision Letter · Decision Letter 1]

15 Jan 2020

PONE-D-19-27232R1

Effect of conservation farming and biochar addition on soil organic carbon quality, nitrogen mineralization, and crop productivity in a light textured Acrisol in the sub-humid tropics

PLOS ONE

Dear Dr Munera-Echeverri,

Thank you for submitting your manuscript to PLOS ONE. After careful consideration, we feel that it has merit but does not fully meet PLOS ONE’s publication criteria as it currently stands. Therefore, we invite you to submit a revised version of the manuscript that addresses the points raised during the review process.

Manuscript is much improved. Please, see comments of the reviewer.

We would appreciate receiving your revised manuscript by Feb 29 2020 11:59PM. To enhance the reproducibility of your results, we recommend that if applicable you deposit your laboratory protocols in protocols.io, where a protocol can be assigned its own identifier (DOI) such that it can be cited independently in the future. For instructions see: http://journals.plos.org/plosone/s/submission-guidelines#loc-laboratory-protocols

We look forward to receiving your revised manuscript.

Kind regards,

Jorge Paz-Ferreiro, Ph.D.

Academic Editor

PLOS ONE

Reviewers' comments:

Reviewer's Responses to Questions

**Comments to the Author**

1. If the authors have adequately addressed your comments raised in a previous round of review and you feel that this manuscript is now acceptable for publication, you may indicate that here to bypass the “Comments to the Author” section, enter your conflict of interest statement in the “Confidential to Editor” section, and submit your "Accept" recommendation.

Reviewer #2: (No Response)

2. Is the manuscript technically sound, and do the data support the conclusions?

Reviewer #2: Yes

3. Has the statistical analysis been performed appropriately and rigorously? 

Reviewer #2: Yes

4. Have the authors made all data underlying the findings in their manuscript fully available?

Reviewer #2: Yes

5. Is the manuscript presented in an intelligible fashion and written in standard English?

Reviewer #2: Yes

6. Review Comments to the Author

Reviewer #2: The authors have done a good job of addressing the comments suggested by both the reviewers. I have one quick comment. As per the suggestion by Reviewer 2, the authors did not include any reference from Sihi et al., 2016, 2017 or 2018 although the author has mentioned that it has been cited. In one instance they have mentioned that the study is nit relatable to Sihi et al., 2017. Please add this reference or refute it with a possible explanation. Other than that I do not have additional comments.

7. PLOS authors have the option to publish the peer review history of their article (what does this mean?). If published, this will include your full peer review and any attached files.

Reviewer #2: No

---

## [Author Response · Author response to Decision Letter 1]

20 Jan 2020

Thanks for the feedback. We had a misunderstanding when handling the three papers by the same main author (Debjani Sihi). We apologize for it. We made a mistake when using the citation generator and forgot to cite the only study that we thought it was directly linked to our paper. From the three papers, only one is related to the effect of soil management practices on soil organic matter in sub-tropical areas (North of India). It is the Sihi et al. (2017) paper:

Sihi, D., Dari, B., Sharma, D. K., Pathak, H., Nain, L., & Sharma, O. P. (2017). Evaluation of soil health in organic vs. conventional farming of basmati rice in North India. Journal of Plant Nutrition and Soil Science, 180(3), 389-406.

In the revised manuscript we have included the reference by Sihi et al. (2017) in line 44, where we discuss the positive effects of increased soil organic matter on water infiltration and soil aggregation. This paper is now number 6 in the reference list.

The other two papers focus on i) temperature sensitivity of soil organic matter decomposition in peat soils and ii) the use of enzyme dynamics in modeling soil organic carbon decomposition and methane emissions. These two topics are not directly linked to our study, although they are about soil organic matter.

1. Sihi, D., Gerber, S., Inglett, P. W., & Inglett, K. S. (2016). Comparing models of microbial–substrate interactions and their response to warming. Biogeosciences, 13(6), 1733-1752.

2. Sihi, D., Inglett, P. W., Gerber, S., & Inglett, K. S. (2018). Rate of warming affects temperature sensitivity of anaerobic peat decomposition and greenhouse gas production. Global change biology, 24(1), e259-e274.

---

## [Editor Report · Decision Letter 2]

23 Jan 2020

Effect of conservation farming and biochar addition on soil organic carbon quality, nitrogen mineralization, and crop productivity in a light textured Acrisol in the sub-humid tropics

PONE-D-19-27232R2

Dear Dr. Munera-Echeverri,

We are pleased to inform you that your manuscript has been judged scientifically suitable for publication and will be formally accepted for publication once it complies with all outstanding technical requirements.

With kind regards,

Jorge Paz-Ferreiro, Ph.D.

Academic Editor

PLOS ONE
---

## [Editor Report · Acceptance letter]

29 Jan 2020

PONE-D-19-27232R2 

Effect of conservation farming and biochar addition on soil organic carbon quality, nitrogen mineralization, and crop productivity in a light textured Acrisol in the sub-humid tropics 

Dear Dr. Munera-Echeverri:

I am pleased to inform you that your manuscript has been deemed suitable for publication in PLOS ONE. Congratulations! Your manuscript is now with our production department. 

With kind regards,

on behalf of

Dr. Jorge Paz-Ferreiro 

Academic Editor

PLOS ONE